# ArgQA: Evaluation of Reasoning Over Elementary Logical Structures in Arguments

## Abstract

As large language models advance in their reasoning capabilities, their adequate evaluation is becoming increasingly important. Existing logical reasoning benchmarks are typically constructed by automatically converting symbolic logic into natural language or curating questions from standardized exams, such as LSAT. However, both synthetic and exam-style questions contain unnatural language, thereby limiting their applicability to real-world contexts. Also, the systematic assessment of reasoning over diverse logical structures remains underexplored. Therefore, we present ArgQA, a novel dataset of 3,807 multiple-choice questions based on authentic arguments from four distinct domains—product reviews, argumentative essays, e-rulemaking comments, and medical research abstracts. Each question is designed to assess the ability to recognize and reconstruct one of three elementary logical structures—linear, convergent, and divergent—whose understanding is a prerequisite to both simple and complex reasoning. Experiments show that even the strongest LLMs still have considerable room for improvement with the overall 9-shot accuracy ranging from 29.2% (Qwen-2) to 61.8% (GPT-o3).

## 1 Introduction

As large language models (LLMs) demonstrate increasingly sophisticated reasoning, adequately assessing their logical reasoning abilities has become crucial for continuing the progress (OpenAI, 2025a;b; Yang et al., 2025). In line with the need, a range of benchmarks have been developed in recent years, typically by automatically converting propositions in symbolic logic into natural language (Saparov & He, 2023; Parmar et al., 2024) or by adopting exams designed for people like LSAT and GMAT (Yu et al., 2020; Zhong et al., 2022). Note, these so-called *logical* reasoning benchmarks focus purely on assessing logical reasoning capabilities. This is distinct from other reasoning benchmarks, which require general or domain-specific knowledge and the ability to reason based on such knowledge. For instance, popular benchmarks HellaSwag (Zellers et al., 2019) and WinoGrande (Sakaguchi et al., 2021) test commonsense reasoning, and GSM8k (Cobbe et al., 2021) and AIME (Patel et al., 2024) assess mathematical reasoning.

For the purpose of evaluating LLMs' reasoning in real-world contexts, however, existing logical reasoning benchmarks are insufficient. For one, they consist of text that bears little resemblance to text in the wild, thereby limiting their relevance to real-world contexts. As one may expect, the disparity is more pronounced for synthetically generated benchmarks, which are comprised of sentences like "Sawyer is a poet." and "Sawyer is either a musician or a poet, but not both." (Qi et al., 2025). However, even those made of exam questions can contain language rarely used in real-life: "Seven directors—A, B, C, D, E, F, and G—serves on the X committee or the Y committee." (Zhong et al., 2022). ReClor (Yu et al., 2020) does have many questions based on realistic arguments, since it adopts LSAT's logical reasoning questions, which are about "arguments as they occur in ordinary language" (LSAC, 2025). However, not all questions follow this style, and the overall linguistic and topical diversity is limited. Also, the systematic assessment of reasoning over diverse logical structures remains underexplored, as questions do not target specific logical structures.

To address these issues, we present ArgumentationQA (ArgQA), a novel dataset of 3,807 multiple-choice questions (MCQs) to assess the ability to recognize and reconstruct elementary logical structures in realistic arguments. As shown in Figure 1, it was constructed based on authentic arguments from four domains—product reviews, argumentative essays, e-rulemaking comments,

**Sentence 1:** If there are no animals in the world, the balance of nature will break down and human beings will die out as well.
**Sentence 2:** Conducting various animal experiments is hazardous to humanity's future and the next generation.

**Which of the following choices is the claim best supported by Sentence 1, while also being the premise best supporting Sentence 2?**
*A: Creatures in the animal kingdom are companionable by nature and remain fundamentally essential to the well-being of humans.* ✓
**B:** Most flowering plants rely on animal pollinators and seed dispersers to reproduce, maintaining food chains supporting humans. ✗
**C:** Drug-resistant pathogens can develop during laboratory animal experiments, potentially breaching containment and sparking lethal pandemics among humans. ✗
**D:** Policymakers ought to eliminate animal testing progressively and encourage the swift creation of ethical, animal-free research techniques. ✗

Argumentative Essays
Question Type: 1.2

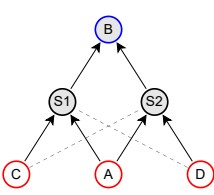

**Sentence 1:** In the subset of patients receiving concurrent chemotherapy, the endpoints again showed no significant differences.
**Sentence 2:** The secondary endpoints did not differ significantly between the HMB/Arg/Gln group and the control group.

**Which of the following choices is the claim best supported by both Sentence 1 and Sentence 2?**
**A:** Numerous participants not receiving chemotherapy significantly reduced their protein consumption throughout the study period, a change that probably obscured any benefits from supplements. ✗
*B: The trial was unable to sufficiently assess whether beta-hydroxy beta-methylbutyrate, glutamine, and arginine can reverse or prevent lean body mass wasting in cancer patients.* ✓
**C:** Strict dose-capping rules designed to limit chemotherapy toxicity produced nearly identical cumulative drug exposure among all participants in that subset, reducing variability in measured endpoints. ✗
**D:** The clinical trial proved statistically underpowered, given that it was able to enroll merely forty-two total patients within the concurrent chemotherapy treatment subset. ✗

Medical Research Abstracts
Question Type: 2.3

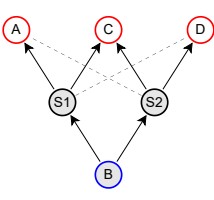

**Sentence 1:** I also support adherence to blue laws that prohibit consumer contact on Sundays.
**Sentence 2:** I conduct no business on Sundays.

**Which of the following choices is the premise best supporting both Sentence 1 and Sentence 2?**
**A:** My blue-law compliance review shows online retail platforms log thousands of Sunday sales, refuting the claim that no business occurs then. ✗
*B: Most individuals typically spend the limited time that remains before the upcoming workweek together with their friends and family on Sundays.* ✓
**C:** Because a voluntary Sunday closure caused no loss of customers or income, it demonstrates that compulsory blue laws forbidding Sunday consumer contact are needless. ✗
**D:** Every Sunday, all of my marketing emails are automatically set up to remain paused from midnight until the following midnight. ✗

e-Rulemaking Comments
Question Type: 3.1

Figure 1: Examples from ARGQA. Each question assesses the ability to understand a particular elementary logical structure. Blue represents the correct option, and red, the wrong options. Gray nodes in the graphs represent the propositions comprising a real argument from the respective domain, which were rephrased by GPT-o3 to make them self-contained, e.g. by replacing pronouns with proper nouns. White nodes represent propositions newly generated with GPT-o3 as the wrong options.

and medical research abstracts—to ensure broad coverage of topics and a spectrum of authorship. The questions were designed around three elementary logical structures—linear, convergent, and divergent, as illustrated in Figure 2—which serve as the basic building blocks of arguments in real-life (Groarke et al., 1997). More specifically, there are nine question types—three types for each of the three elementary logical structures—each of which specifies how the correct and incorrect options are logically linked to the provided context, as shown in Figure 3. Because each incorrect option captures a particular misunderstanding of the logic, additional insights can be gained by analyzing incorrect responses. Lastly, the questions are in multiple-choice format to facilitate an easy integration of ARGQA into the suite of popular MCQ benchmarks that new LLMs are tested on at the time of their release, such as MMLU (Hendrycks et al., 2021) and GSM8k (Cobbe et al., 2021).

With ARGQA, we evaluate the latest LLMs' abilities to understand elementary logical structures across various domains. In particular, we experiment with popular open-source LLMs—Mistral-7B-it-v0.3 (Jiang et al., 2023), Llama-3.1-8B-it (Grattafiori et al., 2024), Qwen-2-7B-it (Yang et al., 2024), and Gemma-7B-it (Team et al., 2024)—as well as strong proprietary models—GPT-o4-mini and GPT-o3 (OpenAI, 2025a). Experimental results show that even the latest models have considerable room for improvement, with the average accuracy ranging from 29.24% (Qwen-2) to 61.81% (GPT-o3). As expected, GPT-o3, the model with one of the strongest reasoning abilities outperforms the rest by a large margin. Among the 7B to 8B parameter models, Gemma achieves the highest overall accuracy of 44.62%, followed by GPT-o4-mini scoring 43.27%. Models exhibit similar performance

on question types sharing the same logical structure, though the propositions comprising them are drastically different. For question types with the same logical structure but with edge direction flipped, models are substantially better at identifying a premise supporting the context, rather than a claim supported by the context. Lastly, medical research abstract is the easiest domain with more clear logical relations, as opposed to argumentative essays with more obscure connections.

## 2 RELATED WORKS

Table 1: Comparison of ARGQA with existing logical reasoning datasets. [†]: Product reviews, argumentative essays, medical research abstracts, and e-Rulemaking comments. [‡]: The context and correct option are a real argument from the source domain paraphrased to be self-contained. Each incorrect option was generated to be logically linked to the context, but to form a logical structure different from the one specified in the question text. [*]: Entailment/Contradiction/Neutral/Paradox.

| Dataset | Size | Source Text | Construction Method | Text Structure | Task |
|---|---|---|---|---|---|
| CLUTRR | 6k | N/A | Machine-Generated | Synthetic Story | T/F |
| RuleTaker | 500k | N/A | Machine-Generated | Set of Propositions | T/F |
| ProofWriter | 500k | N/A | Machine-Generated | Set of Propositions | T/F with Proof |
| LogicNLI | 20k | N/A | Machine-Generated | Set of Propositions | E/C/N/P[*] |
| SimpleLogic | 560k | N/A | Machine-Generated | Set of Propositions | T/F |
| PrOntoQA | 7.9k | N/A | Machine-Generated | Set of Propositions | T/F with Proof |
| LogicBench | 2k | N/A | Machine-Generated | Synthetic Story | Y/N and MC |
| ProverQA | 1.5k | N/A | Machine-Generated | Set of Propositions | T/F with Proof |
| FOLIO | 1.4k | N/A | Human-Written | Set of Propositions | T/F/Unknown |
| LogiQA | 8.6k | NCSE | Curated (Human-Written) | Mixed | MC |
| ReClor | 6.1k | GMAT, LSAT | Curated (Human-Written) | Mixed | MC |
| AR-LSAT | 2k | LSAT | Curated (Human-Written) | Mixed | MC |
| **ARGQA (ours)** | **3.8k** | **Various**[†] | **Paraphrased or Generated**[‡] | **Argument** | **MC** |

**Logical Reasoning Datasets.** Given the growing need for LLMs to perform sophisticated logical reasoning, numerous datasets focusing specifically on assessing their logical reasoning abilities have been developed in recent years, as shown in Table 1. One group of logical reasoning datasets adopt questions from exams designed for people: LogiQA (Liu et al., 2021) and LogiQA 2.0 (Liu et al., 2023) consist of questions from National Civil Servants Exam (NCSE) of China, whereas ReClor (Yu et al., 2020) and AR-LSAT (Zhong et al., 2022) incorporate US-based standardized tests: LSAT and GMAT for the former, and LSAT for the latter. These datasets cover various types of questions, including drawing a conclusion from a set of premises and inferring an event from a synthetic scenario like "Seven directors—A, B, C, D, E, F, and G—serves on the X committee or the Y committee."

The other group comprises synthetic sentences generated with rules and templates or using LLMs. These datasets were typically generated based on propositions in symbolic logic, where logical structures among the propositions are easier to control: With such high degree of control, questions were carefully designed to assess the ability to identify logical paths linking provided propositions and propositions to be verified. In RuleTaker (Clark et al., 2021), ProofWriter (Tafjord et al., 2021), LogicNLI (Tian et al., 2021), and SimpleLogic (Zhang et al., 2023), the questions are formulated as confirming new facts using provided facts and logical rules. PrOntoQA (Saparov & He, 2023) and PrOntoQA-OOD (Saparov et al., 2023) only focus on *modus ponens* deduction rule. ProverQA (Qi et al., 2025) makes the problem more difficult with so-called distractors, which are propositions unnecessary for the proof. Unlike other datasets in this group, FOLIO (Han et al., 2024) was manually written based on real world knowledge available in Wikipedia. Lastly, CLUTRR (Sinha et al., 2019) and LogicBench (Parmar et al., 2024) assess logical reasoning in the context of synthetic scenarios, rather than sets of propositions.

ARGQA complements these datasets with questions based on real arguments from various domains, carefully designed to assess the ability to understand elementary logical structures. This preserves transferability to real-world contexts, while allowing more fine-grained analyses of models with respect to different domains and logical structures.

**Argument Mining Datasets.** As will be discussed in Section 3.1, an argument in natural language is a set of premises supporting a claim, where the premises and the claim are propositions that are either true or false. Argument mining is a task of identifying and extracting arguments in text, which can assist deeper comprehension and critical evaluation, as well as the generation of relevant text, such as a counter-argument. To support research in this direction, many argument mining datasets have been developed over the years. Because the structure and style of writing vary from one domain to another, a dataset typically targets a single domain, such as news articles (e.g. Eckle-Kohler et al. 2015; Al-Khatib et al. 2016; Ein-Dor et al. 2020), legal documents (e.g. Poudyal et al. 2020; Grundler et al. 2022), political debates (e.g. Haddadan et al. 2019; Visser et al. 2019; Hautli-Janisz et al. 2022), student essays (e.g. Stab & Gurevych 2017; Alhindi & Ghosh 2021; Schaller et al. 2024), and user-generated content online (e.g. Boltužić & Šnajder 2014; Habernal & Gurevych 2017; Bhatti et al. 2021). In addition to the differences in domain, the authors may be driven by distinct goals, leading to different annotated components (e.g. different types of premises) and relations (e.g. different types of support relations). Also, the task itself can be posed as a structured prediction problem (e.g. input: an entire document, output: a directed graph representing the logical structure) or a binary classification problem (e.g. input: an ordered pair of propositions, output: a binary verdict on whether the first proposition supports the second), among others. Due to the limited standardization, it can be difficult to use these datasets for assessing the logical reasoning abilities of models. Also, analyzing models' mistakes to gain additional insights is cumbersome. To the best of our knowledge, ARGQA is the first dataset in a standardized format that has been designed to evaluate the capacity to recognize elementary logical structures in real arguments and analyze mistakes in a convenient manner.

# 3 THE ARGUMENTATIONQA (ARGQA) DATASET

## 3.1 THEORETICAL BACKGROUND

**Elementary Logical Structures.** *Argumentation* refers to the process of constructing a natural language argument, a set of propositions (*premises*), and a target proposition (*claim*) logically supported by the premises (Stede & Schneider, 2018). With natural language arguments, unlike in formal-logic, the same sentence can be interpreted in many different ways, and their logical relationships are often not as straight-forward as, say, deductive inference. Also, some of the premises are typically left implicit—resulting in arguments known as *enthymemes* (Blair & Johnson, 1987; Walton, 2009)—because it is prohibitive to list all premises needed to support a given claim. Such fluidity of language and practical constraints limit the applicability of formal logic, whose rules are clean yet rigid. Thus, argumentation theory aims to bridge the gap between formal logic and everyday reasoning with additional flexibility from relatively under-specified logical relations like "support." Through the study of reasoning in natural language, argumentation theorists have identified a few elementary logical structures that serve as the building blocks for all arguments (Groarke et al., 1997; Rahwan, 2008; Lawrence & Reed, 2019). Here are three common elementary logical structures in practical argumentation (Examples can be found in Figure 2):

1. **Linear Structure.** A proposition supports another proposition, which in turn, supports yet another proposition.[1]

2. **Convergent Structure.** Multiple propositions independently support a proposition.

3. **Divergent Structure.** A proposition independently supports multiple propositions.

There is a fourth type of elementary logical structure, namely the *linked* structure. This structure is similar to the convergent structure in that multiple propositions support a proposition. In this case, however, the supporting propositions work collectively, thus all supporting propositions are needed for the argument to work. We exclude this structure from our work, due to its infrequent use in practice, which limits its relative significance and poses challenges for data collection.

**Question Types.** Recognizing or constructing elementary logical structures—which serve as the building blocks for both simple and complex logical structures—is a prerequisite for strong logical reasoning. One way to assess this ability in a standardized format is by designing MCQs that require selecting a proposition which, when combined with two provided propositions, forms a target

---

[1]Here, the second proposition is a *claim* with respect to the first, and *premise* in relation to the third.

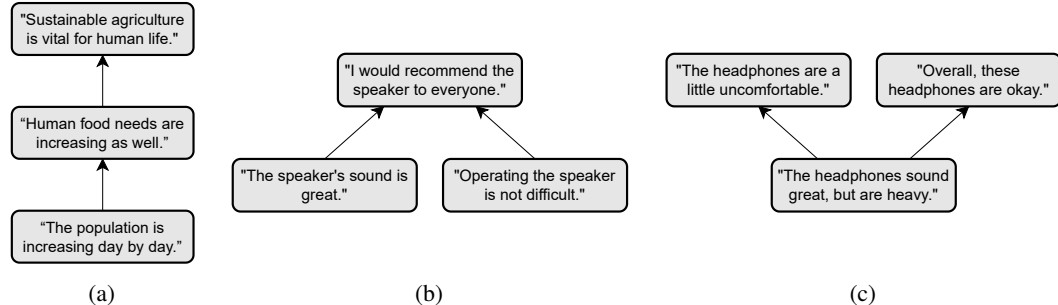

Figure 2: Examples of three elementary logical structures found in practical argumentation: (a) linear, (b) convergent, and (c) divergent structure. Directed edges represent support relations.

elementary logical structure. Since each structure consists of three propositions, we define three question types for each structure for a total of nine question types, as presented in Figure 3.

The logical distractors are designed so that they are logically related to one or both of the context propositions. This results in their forming various logical structures with the context, requiring the capability to correctly recognize the specific logical structure described in the question. In addition, this naturally prevents the correct option from being the only option that is logically related to the context—or, even worse, the only option that is topically relevant. Both cases can be easily exploited, potentially hindering a proper evaluation of the logical reasoning abilities.

Note, we assume the transitivity of support relations. That is, if proposition $a$ supports proposition $b$, which in turn supports proposition $c$, then proposition $a$ also supports proposition $c$. This is typically, though not necessarily, the case in real arguments. As a side-effect, this limits the ways in which distractors are related to the context. For instance, in question type 3.2, a distractor cannot be supported by S1, because it would form the correct structure with the context. More over, it cannot be supported by S2, because that would mean it is also supported by S1—by transitivity—which forms the correct structure. Thus, the only possible logical relations for distractors are supporting S1 or S2.

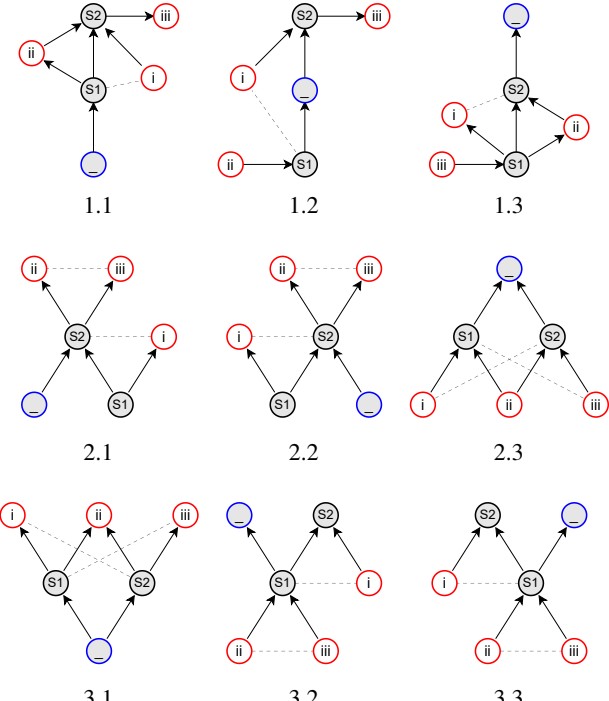

Figure 3: Question types and their logical structures. Directed edges represent support relations, and dashed edges represent the explicit lack of support relations in either direction. Grey nodes constitute the target elementary logical structure formed by the context *S1*, *S2*, and the correct option _ in blue. The three nodes in red are incorrect options, which are logical distractors related to the context in a structure different from what is described in the question.

## 3.2 DATASET CONSTRUCTION

We aim for questions based on real arguments to assess the capacity to understand elementary logical structures. Following the aforementioned question design, we construct the ARGQA dataset through a multi-stage pipeline, with each step optimized through pilot experiments:

**Step 1: Source Selection.** We select argument mining datasets to acquire raw text from diverse domains, along with their corresponding logical relation annotations. We leverage publicly available argument mining datasets, which annotate documents with logical structures. However, we exclude those with synthetic text (e.g. Peldszus & Stede 2015), with an annotation scheme disallowing one or more of the three elementary logical structures (e.g. Poudyal et al. 2020), and without enough context to make propositions interpretable (e.g. Hautli-Janisz et al. 2022). From over 20 datasets we considered, only the following meet the criteria: (1) AAEC2 (Stab & Gurevych, 2014) consists of argumentative essays by students on contentious issues, often using implicit reasoning allowing for multiple interpretations; (2) CDCP (Park & Cardie, 2018) is composed of user comments on e-Rulemaking, presenting opinions and related experiences regarding consumer debt collection practices; (3) AbstRCT (Mayer et al., 2020) features abstracts of Randomized Controlled Trials (RCT) from the MEDLINE database, often displaying clear and concise logic; and (4) AM$^2$ (Chen et al., 2022) is a collection of product reviews from Amazon, characterized by grammatical flexibility.

**Step 2: Triplet Extraction.** From each dataset, we extract triplets of propositions that form elementary logical structures of our interest: *linear*, *convergent*, and *divergent*. For this, we first construct directed graphs of logical relations from the annotations, then extract triplets of propositions tagged with the elementary logical structure they form. The triplets are then filtered to prevent semantic overlaps while balancing the number of triplets across the logical structures: All triplets in the divergent structure—the most rare one—are selected, then the same number of triplets for each of linear and convergent structure are selected, with each proposition appearing in one triplet only.

**Step 3: Proposition Paraphrasing.** We paraphrase each proposition to make them self-contained. Argument mining datasets typically keep the original text, thus some propositions cannot be interpreted without the context. However, because our questions contain triplets without the context, they need to be self-contained. Thus, we use GPT-o3 to paraphrase each proposition to a self-contained and grammatically complete sentence, through co-reference resolution and sentence completion using the original context. For example, *"JUNK!"* is paraphrased as *"These headphones are junk!"* based on the context of the product review in which it appears. (See Table 7 for the prompt.)

**Step 4: Triplet Deduplication.** We filter triplets comprised of semantically similar propositions. We do so by measuring the cosine similarity between the SBERT embeddings (Reimers & Gurevych, 2019) of all pairs of propositions within each triplet. When the similarity is above a threshold, the triplet is discarded. This is to exclude triplets where semantically identical or similar propositions have been wrongly annotated as in support relations.

**Step 5: Distractor Generation.** For each triplet, we construct three *context-options* pairs by splitting the triplet into context—two propositions—and one correct option in three ways. Each pair is tagged with the corresponding question type. Then, for each pair, we generate three logical distractors with GPT-o3 by feeding in the context and a description of the logical relation between the context and the proposition to be generated, specified by the question type. These distractors constitute the incorrect options and are added to the respective *context-options* pairs. (See Table 8 for the prompt.)

**Step 6: Option Rephrasing.** For each *context-options* pair, we rephrase both correct and incorrect options to similar lengths. This is to prevent potential differences in lengths revealing the correct option. For this, we compute the median word count across the four options and paraphrase each option to have its length fall within two words of the median. (See Table 9 for the prompt.)

**Step 7: Question Construction.** Finally, we construct a question for each *context-options* pair by combining it with a pre-written question describing the logical relationship between the context and the correct option, such as "Which of the following choices is the premise best supporting Sentence 1?" for question type 1.1. (See Table 4 for the full list of question text.) Each question is composed of two context propositions, one question text, and four options—of which one is correct—in a randomized order. Note, each question is tagged with domain, as well as the question type.

### 3.3 RESULTING DATASET

An summarized in Table 2, ARGQA consists of 3,807 MCQs, evenly split across the nine question types, and in turn the three elementary logical structures. For each domain, nine questions are reserved as the development set used as examples during few-shot experiments. The remainder are randomly split into 10% validation and 90% test set. Domains are distributed as follows: 1,620

questions on product reviews (42.6%), 1,350 on argumentative essays (35.5%), 567 on e-Rulemaking comments (14.9%), and 270 on medical research abstracts (7.1%). Interestingly, their ranking by length is the opposite: medical research abstracts has the longest context and option propositions, followed by e-Rulemaking comments, argumentative essays, and product reviews. The same goes for ordering by linguistic diversity as measured by vocabulary size per instance.

To confirm the quality, we manually analyzed 108 validation instances randomly selected through stratified sampling, i.e., 27 per domain. 93.8% of the options are in the correct relation to the context as specified by the question type. The common error patterns differ for the correct and the incorrect options. For correct options, the lack of explicit context often obscures the logic. For instance, the relation between two propositions each discussing a company's and a student's perspective is not as clear without knowing that the student is an intern. For incorrect options, it is common to have a reason embedded in the option, e.g. "*B, because A.*" Then, it is not supported by a context proposition stating *A*—though it was supposed to—since *A* is already stated in the option.

Table 2: Overview of ARGQA. Source arguments were collected from argument mining datasets across four domains for a wide range of topics and writing styles. Each domain-split contains the same number of questions for each of the nine question types. For instance, the validation set for e-Rulemaking comments comprises six questions per question type, totaling 54. ∗: number of words. ♣: Chen et al. (2022). ♦: Stab & Gurevych (2014). ♥: Park & Cardie (2018). ♠: Mayer et al. (2020).

| Domain | Source Dataset | # of Instances | | | | Vocab* | Avg. Sentence Len. | |
| | | Dev | Val | Test | Total | | Context* | Option* |
|---|---|---|---|---|---|---|---|---|
| Product Reviews | AM$^2$ ♠ | 9 | 162 | 1,449 | 1,620 | 8,616 | 11.6 | 15.7 |
| Argumentative Essays | AAEC2 ♦ | 9 | 135 | 1,206 | 1,350 | 11,465 | 16.8 | 20.3 |
| e-Rulemaking Comments | CDCP ♥ | 9 | 54 | 504 | 567 | 6,211 | 18.0 | 21.1 |
| Medical Research Abstracts | AbstRCT ♠ | 9 | 27 | 234 | 270 | 4,640 | 26.9 | 24.4 |
| **ARGQA** | | 36 | 378 | 3,393 | 3,807 | 18,452 | 15.5 | 18.8 |

# 4 EXPERIMENTS

## 4.1 EXPERIMENTAL SETUP

We use ARGQA to assess the logical reasoning abilities of popular open-source LLMs—Mistral-7B-Instruct-v0.3 (Jiang et al., 2023), Llama-3.1-8B-Instruct (Grattafiori et al., 2024), Qwen-2-7B-Instruct (Yang et al., 2024), and Gemma-7B-Instruct (Team et al., 2024)—as well as strong proprietary models—GPT-o4-mini and GPT-o3 (OpenAI, 2025a). We focus on 7B to 8B parameter models for a fair comparison. However, GPT-o3 is also included as a representative of the LLMs with the strongest reasoning capabilities; among the available GPT models we pilot-tested, GPT-o3 performed the best. Experiments were conducted on the held-out test set in zero-shot and nine-shot settings, using the development set as examples of the nine question types. The performance was measured in accuracy, i.e., the percentage of correct responses. We adapted a multiple-choice prompting protocol from MMLU-Pro (Wang et al., 2024), instructing models to return answers in the format "the answer is (X)", and automatically extracting the answer using regular expressions. The exact wording (See Figure 10 for the prompt), as well as maximum output length (32 tokens) and repetition penalty (1.0) were finalized based on the validation performance. Lastly, open-source models were evaluated with greedy decoding, with temperature set to 0, for maximally deterministic responses.

## 4.2 RESULTS & ANALYSIS

As presented in Table 3[2], the overall performance of the LLMs measured in accuracy on the ARGQA test set for all domains ranges from 29.24% (Qwen-2) to 61.81% (GPT-o3), with GPT-o3 considerably outperforming the rest of the models. This is in line with expectations, since GPT-o3 is a substantially larger model specializing in logical reasoning. Among the other models—which all have 7B to 8B parameters—Gemma is generally the best performer, followed by GPT-o4-mini. Qwen-2 and Mistral rank the lowest with only a modest 4% improvement from the random baseline of 25%,

---

[2]See Table 6 for 0-shot results. As expected, the performance generally degrades without examples.

Table 3: Nine-shot performance of LLMs on the ARGQA test set for all domains. Each cell reports accuracy (%), the percentage of questions that were correctly answered. Open-source models are evaluated with greedy decoding, temperature set to 0, for maximally deterministic responses. LLMs show considerable room for improvement, with GPT-o3 substantially outperforming the rest.

| Logical Structure | Mistral | LLaMA-3.1 | Qwen-2 | Gemma | GPT-o4-mini | GPT-o3 | Avg. |
|---|---|---|---|---|---|---|---|
| Linear | 30.77 | 31.92 | 24.84 | 36.69 | 41.47 | 55.71 | 36.90 |
| Q-type: 1.1 | 37.67 | 43.50 | 29.97 | 46.95 | 54.38 | 59.95 | 45.40 |
| Q-type: 1.2 | 29.97 | 33.16 | 27.85 | 41.38 | 43.24 | 65.00 | 40.10 |
| Q-type: 1.3 | 24.67 | 19.10 | 16.71 | 21.75 | 26.79 | 42.18 | 25.20 |
| Convergent | 30.15 | 41.03 | 29.97 | 52.96 | 50.22 | 64.99 | 44.89 |
| Q-type: 2.1 | 34.75 | 51.46 | 36.87 | 63.13 | 60.74 | 68.97 | 52.65 |
| Q-type: 2.2 | 32.89 | 44.56 | 35.28 | 61.01 | 58.09 | 64.72 | 49.43 |
| Q-type: 2.3 | 22.81 | 27.06 | 17.77 | 34.75 | 31.83 | 61.27 | 32.58 |
| Divergent | 26.97 | 31.48 | 32.89 | 44.21 | 38.11 | 64.72 | 39.73 |
| Q-type: 3.1 | 48.01 | 58.89 | 46.68 | 74.54 | 75.07 | 80.64 | 63.97 |
| Q-type: 3.2 | 16.98 | 18.04 | 27.85 | 29.18 | 19.10 | 57.03 | 28.03 |
| Q-type: 3.3 | 15.92 | 17.51 | 24.14 | 28.91 | 20.16 | 56.50 | 27.19 |
| All (*Q-type Avg.*) | 29.30 | 34.81 | 29.24 | 44.62 | 43.27 | 61.81 | 40.51 |

which corresponds to randomly selecting the correct answer from four options. Overall, these results indicate that LLMs currently have insufficient abilities to recognize elementary logical structures, even those comprised of only three propositions. In other words, understanding logical structures, no matter how simple, remains a challenging problem to modern LLMs.

Analyses of the results with respect to the logical structures provide additional insights. For instance, consider Q-types 2.1 and 2.2, which share the same logical structure, but consist of different propositions: Of the six propositions making up a question, one is the same (a context proposition), two are paraphrases and play different roles (one context proposition and the correct option, and vice versa for the other Q-type), and three are entirely distinct. (Figure 6 shows how different the questions are.) Even with the considerable differences in the propositions forming the question, the performance on these question types is quite similar across the models. On the other hand, the performance is drastically different on Q-type 2.3, which shares the same type of similarities in propositions, but has a different logical structure. This suggests that the logical structure has more impact on the model response than the particular propositions forming the structure, which aligns with the goal of assessing logical reasoning with an emphasis on the structure. The same trend can be observed for Q-types 3.2 and 3.3, which also share the same logical structure but different propositions.

Also, consider Q-types 2.3 and 3.1. The logical structures for these Q-types are mirror-images of each other, i.e., they are identical, except the edges point in the opposite direction. Unlike the cases where the logical structures are the same, the performance varies drastically between the pair, with the models averaging 32.58% for the former, and 63.97% for the latter. A similar pattern can be observed between another pair with the same logical structure but flipped edges: Q-types 1.1 and 1.3. One plausible explanation is that the models are better at identifying a premise supporting a given claim(s), as opposed to recognizing a claim supported by a given premise(s). This observation is also consistent with the fact that the Q-types with the highest performance reported—Q-types 3.1, 2.1, 2.2, and 1.1—all require identifying a premise supporting claim(s) provided as context.

Among the domains, argumentative essays are the most difficult for the models, with accuracies averaging 33.78%. Arguments on contentious topics typically involves so-called *defeasible* reasoning in argumentation theory, which can be challenged in several ways, e.g. *rebutting* the claim with a different reason or *undercutting* the logical connection. For instance, in Example 1 in Figure 1, one may undercut the link between sentence 2 and option A by saying that animals used for experiments constitute only a negligibly small portion of all animals. In this way, logical connections are less concrete in this domain. On the other hand, the models perform best on medical research abstracts, with an average accuracy of 50.0%. This is partly because abstracts summarizing research papers tend to have clearer logical relations. For instance, the supported proposition can simply be generalizing the propositions that support it with more specific cases, as shown in Example 2 in Figure 1.

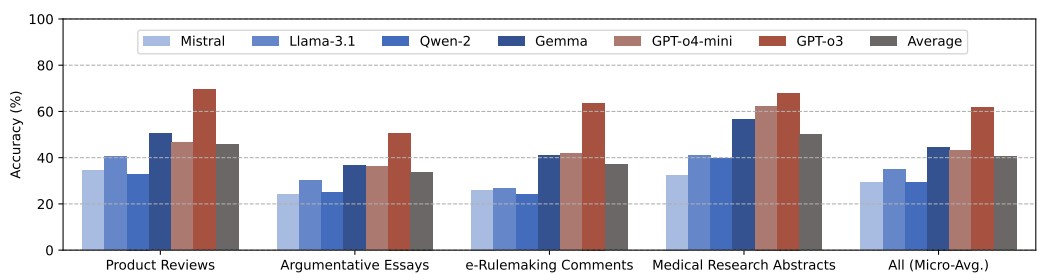

Figure 4: Nine-shot performance of LLMs on the ARGQA test set for each domain. On average, as shown in gray bars, LLMs perform best on medical research abstracts, followed by product reviews, e-Rulemaking comments and argumentative essays.

The selection rates for the logical distractors are presented in Figure 5. Note that distractors i, ii, and iii can be any of the options A, B, C, or D in a given question, since the order is randomized. Thus, patterns shown in this figure do not result from locational bias. One trend is the models' strong preference toward the distractor linked to both context propostions. Distractor ii in Q-types 1.1, 1.3, 2.3, and 3.1 is one such distractor, and we can observe a marked inclination toward choosing it. It seems that when models fail to identify a proposition logically related to the context in a way described in the question text, they favor the proposition related to more context propositions. This does not reflect a unconditional predisposition to select the proposition associated with both context propositions, because if that were the case, models would have performed much better on Q-type 1.2, where the only proposition related to both context propositions is the correct option. Regarding Q-types with shared structures—Q-types 2.1 and 2.2, as well as 3.2 and 3.3—we not only observe similar performance, as previously discussed, but also similar distractor selection patterns; the stacked bars are near identical for these pairs across the models. This again confirms that the logical structure has a stronger impact on the choice than the particular propositions comprising the structure, which is desirable for evaluating reasoning over various logical structures.

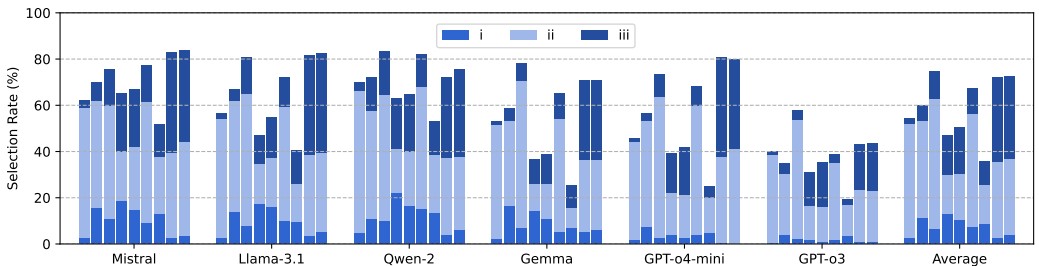

Figure 5: Selection rates for the distractors. Each group of nine stacked bars represent the percentages of selecting the three distractors for the nine question types. The stacks do not add up to 100%, because the percentage of selecting the correct option is not shown. Note, generalized claims about a particular distractor number, e.g. distractor i, cannot be made based on this figure, because the distractors' logical relationship to the context is different for each question type, as shown in Figure 3.

## 5 CONCLUSIONS

In this work, we introduce ARGQA, a novel dataset designed to evaluate logical reasoning abilities over elementary logical structures: linear, convergent, and divergent. By moving beyond synthetic and exam-style questions to realistic arguments from various domains, ARGQA addresses the critical issue of limited applicability suffered by existing logical reasoning benchmarks. Also, the systematic design of question types enables convenient analyses of error patterns with respect to each elementary logical structure. Experiments on the latest LLMs reveal that even the strongest reasoning models like GPT-o3 have considerable room for improvement. With its standardized format, we hope ARGQA becomes a valuable resource for assessing LLMs' reasoning capability.

## 6 ETHICS STATEMENT

ARGQA is constructed by leveraging publicly available argument-mining datasets (AM$^2$, CDCP, AAEC2 and AbstRCT) to paraphrase and generate new propositions. Our project did not involve any human participants, and no private or personally identifying information was collected or published.

## 7 REPRODUCIBILITY STATEMENT

Included with our submission are all the code for dataset construction and experiments, along with the data. This is to ensure the transparency in data construction, verification of the experimental process, and reproducibility of the results. Upon acceptance, we plan to formally release the code and the dataset via an open source repository.

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

# A APPENDIX

## A.1 ADDITIONAL DETAILS

Table 4: Question text for each question type.

| Type | Question Text |
|------|---------------|
| 1.1 | Which of the following choices is the premise best supporting Sentence 1? |
| 1.2 | Which of the following choices is the claim best supported by Sentence 1, while also being the premise best supporting Sentence 2? |
| 1.3 | Which of the following choices is the claim best supported by Sentence 2? |
| 2.1 | Which of the following choices is the premise best supporting Sentence 2? |
| 2.2 | Which of the following choices is the premise best supporting Sentence 2? |
| 2.3 | Which of the following choices is the claim best supported by both Sentence 1 and Sentence 2? |
| 3.1 | Which of the following choices is the premise best supporting both Sentence 1 and Sentence 2? |
| 3.2 | Which of the following choices is the claim best supported by Sentence 1? |
| 3.3 | Which of the following choices is the claim best supported by Sentence 1? |

Table 5: Content of each ARGQA instance. The order of the four answer choice options is pre-randomized in the released file, ensuring that solvers cannot benefit from a positional bias.

| Field | Type | Description |
|-------|------|-------------|
| docID | integer | Index that links the item to its source document in the original argument mining corpus. |
| instanceID | string | Globally unique identifier formed from the split label (dev, val, test) and a counter. |
| structure | string | The logical structure label, one of lin, conv, div for linear, convergent, or divergent arguments. |
| q_type | string | Question type code, for example "1.1" for proposition prediction. |
| context | array of 2 strings | Two sentences that make up the argument fragment shown to the solver. |
| choices | array of 4 objects | Each object has text and type. |
|   text | string | Answer text shown to the solver. |
|   type | string | Categorical label where _ marks the single gold answer whose reasoning chain matches the target structure, and i, ii, iii mark other distractor subtypes (for example simple backward, complex forward, complex linear). |

Table 6: Zero-shot performance of LLMs on the ARGQA test set for all domains. Each cell reports accuracy (%), the percentage of questions that were correctly answered. Greedy decoding, with temperature set to 0, was used to minimize variance. All LLMs tested show considerable room for improvement, with GPT-o3 substantially outperforming the rest.

| Logical Structure | Mistral | LLaMA-3.1 | Qwen-2 | Gemma | GPT-o4-mini | GPT-o3 | Avg. |
|---|---|---|---|---|---|---|---|
| Linear | 26.79 | 27.23 | 24.14 | 27.76 | 39.79 | 46.42 | 32.02 |
| Q-Type: 1.1 | 27.59 | 36.60 | 30.50 | 38.73 | 50.66 | 51.99 | 39.35 |
| Q-Type: 1.2 | 32.10 | 22.55 | 24.40 | 23.34 | 42.71 | 57.29 | 33.73 |
| Q-Type: 1.3 | 20.69 | 22.55 | 17.51 | 21.22 | 25.99 | 29.97 | 22.99 |
| Convergent | 31.04 | 38.02 | 34.30 | 39.88 | 49.43 | 49.87 | 40.42 |
| Q-Type: 2.1 | 32.10 | 40.58 | 43.50 | 45.89 | 58.36 | 56.23 | 46.11 |
| Q-Type: 2.2 | 30.77 | 41.91 | 40.58 | 44.56 | 53.58 | 53.85 | 44.21 |
| Q-Type: 2.3 | 30.24 | 31.56 | 18.83 | 29.18 | 36.34 | 39.52 | 30.95 |
| Divergent | 30.86 | 31.12 | 33.42 | 33.24 | 38.37 | 42.62 | 34.94 |
| Q-Type: 3.1 | 42.18 | 48.54 | 53.58 | 57.82 | 68.70 | 68.97 | 56.63 |
| Q-Type: 3.2 | 23.08 | 22.81 | 24.14 | 20.69 | 22.55 | 29.97 | 23.87 |
| Q-Type: 3.3 | 27.32 | 22.02 | 22.55 | 21.22 | 23.87 | 28.91 | 24.32 |
| All (*Micro-Avg.*) | 29.56 | 32.12 | 30.62 | 33.63 | 42.53 | 46.30 | 35.79 |

## A.2 EXAMPLE QUESTIONS

**Sentence 1:** Wiggling the headphone jack caused both speakers to become intermittent.
**Sentence 2:** The QC-25 headset proved to be a disappointment.

**Which of the following choices is the premise best supporting Sentence 2?**
*A: After fewer than three minutes, the left speaker completely ceased to operate.* ✓
**B:** I sent the QC-25 headset back to the store and obtained a complete refund. ✗
**C:** A new cable instantly restored perfect sound, proving the QC-25 itself was blameless. ✗
**D:** QC-25's poor performance drove our audio-engineering club to schedule a seminar on noise-cancellation design flaws. ✗

Product Reviews
Question Type: 2.1

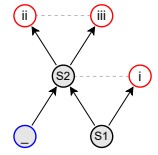

**Sentence 1:** The left speaker stopped working after less than three minutes.
**Sentence 2:** The QC-25 headset proved to be a disappointment.

**Which of the following choices is the premise best supporting Sentence 2?**
**A:** I asked the retailer to provide me with a complete refund. ✗
**B:** I quickly returned defective headphones, bought a QC-25 that has long performed flawlessly. ✗
**C:** The engineering group launched an internal probe into the product's acoustic flaws. ✗
*D: Jiggling the headphone jack caused both of the speakers to cut out intermittently.* ✓

Product Reviews
Question Type: 2.2

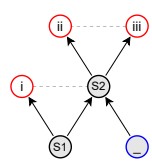

Figure 6: Examples with Equal Structure but Different Propositions

## A.3 EXAMPLE PROMPTS

### Table 7: Prompt for Proposition Paraphrasing for AM[2]

**Developer Prompt:**
# Identity
You are an expert editor that rewrites the sentence to become grammatically complete if necessary.

# Instructions
Your task is to rewrite the original sentence, such that:
1. The generated sentence becomes a grammatically complete sentence that can stand on its own.
2. The generated sentence preserves the original meaning without adding new details or elaboration.
3. All pronouns (e.g., "he", "she", "it", "they") are replaced with appropriate proper nouns or clear references from the given Context.
4. Remove connectors (e.g., "because", "but", "so", "in order that") if necessary.
5. The original point of view (e.g., first-person) must be preserved.
6. Only output a single rewritten sentence. Do not include explanations, formatting, or additional commentary.

# Examples
<product_review id="example-1">
Context: The speakers aren't even oriented around your ears, they're cockeyed.
Original Sentence: they're cockeyed.
</product_review>
<assistant_response id="example-1">
The speakers are cockeyed.
</assistant_response>

<product_review id="example-2">
Context: I hate this headset. Connection is terrible.
Original Sentence: I hate this headset.
</product_review>
<assistant_response id="example-2">
I hate this headset.
</assistant_response>

<product_review id="example-3">
Context: Spend a bit more money and get something better. I will, I have to now.
Original Sentence: I have to now.
</product_review>
<assistant_response id="example-3">
I have to purchase better quality headphones now.
</assistant_response>

<product_review id="example-4">
Context: Had to return it because the sound quality was not good.
Original Sentence: because the sound quality was not good.
</product_review>
<assistant_response id="example-4">
The sound quality was not good.
</assistant_response>

**User Prompt:**
Context: {full_context}
Original Sentence: {original_sentence}
Completed Sentence:

Table 8: Prompt for Generating Distractors for AM[2]

---

**Developer Prompt:**

# Identity

You are an expert natural language logician. Your task is to generate a sentence that serves as the logical bridge between two given sentences. Given two sentences, A and B, your task is to generate a new sentence C, such that A "is a reason" for C, and C "is a reason" for B (A -> C -> B). You must make sure neither reverse relation holds. B must NOT be a reason for C, and C must NOT be a reason for A.

# Instructions

* Sentence A must be a reason for sentence C in natural language, and sentence C must simultaneously be a reason for sentence B in natural language. Imagine the full sequence as "[Sentence A]. Because of this reason, [Sentence C]. Because of this reason, [Sentence B]."
* The relation must only go one way. Sentence B must NOT be a reason for sentence C, and sentence C must NOT be a reason for sentence A. If sentence B is a reason for sentence C, or sentence C is a reason for sentence A, your output is invalid.
* Sentence C must be distinct in meaning from sentence A or B. It must contain a new proposition without repetition from sentence A or B.
* Sentence C must be a sentence that can stand on its own. It must not have any unresolved references like pronouns that rely on sentence A or B (e.g., "it", "they", "them").
* Your response must be the single generated sentence C, with no additional formatting or explanation.

# Examples

<sentence id="good-example-1">
Sentence A: The Bluetooth signal comes solely from the right earphone.
Sentence B: The issue with this Bluetooth headset can be easily overcome.
</sentence>
<assistant_response id="good-example-1">
The Bluetooth headset only has skip issues when the cell phone is in the left pocket.
</assistant_response>

<sentence id="good-example-2">
Sentence A: The instruction booklet was easy to read and understand.
Sentence B: I am glad I bought these headphones.
</sentence>
<assistant_response id="good-example-2">
I had no trouble getting the headphones out and figuring out how to use them.
</assistant_response>

<sentence id="good-example-3">
Sentence A: I am very disappointed.
Sentence B: Spend the extra money.
</sentence>
<assistant_response id="good-example-3">
Go with the more expensive alternative.
</assistant_response>

<sentence id="bad-example-3">
Sentence A: One speaker went out in less than one month.
Sentence B: The second speaker failed soon after.
</sentence>
<assistant_response id="bad-example-3">
Don't waste your money on them.
</assistant_response>
This is a bad example because both A and B support C. A, B are both reasons for C, which is unacceptable. Additionally, this example uses an unresolved reference ("them"), which is also unacceptable.

---

**User Prompt:**

Sentence A: {first_sentence}
Sentence B: {second_sentence}

---

Table 9: Prompt for Option Rephrasing for AAEC2

**Developer Prompt:**
# Identity
You are an expert editor who rewrites sentences to precisely match a target word length.

# Instructions
Your task is to rewrite the Original Sentence such that:
* The rewritten sentence conveys exactly the same meaning. No information must be added, removed, or altered.
* The rewritten sentence must contain about the same number of words as the target, within a two-word tolerance. Verify the word count of your rewritten sentence and revise it until it matches that target. However, do not add or remove any content in order to meet this word count — preserving the original meaning is more important than exact length.
* The original point of view and tense must remain unchanged. You must NOT add any new unresolved references like pronouns.
* Your response must be exactly one grammatically complete and independent sentence. Do not split it into multiple sentences.
* Output only the rewritten sentence without any explanations, formatting, or additional commentary.

# Examples
<sentence id="good-example-1">
Sentence: Budget constraints make hiring temporary replacements financially impossible for management.
Number of words targeted: 30
</sentence>
<assistant_response id="good-example-1">
Because the already strained budget leaves no available funds, management finds that hiring short-term replacement workers is entirely out of reach financially, rendering any temporary staffing plan impossible.
</assistant_response>

<sentence id="good-example-2">
Sentence: Analyses of thousands of successful career trajectories show that bold self-direction paired with meticulous preparation—not mere optimism that any gamble will succeed—is the chief driver of achievement, thereby challenging the claim that simply taking chances and believing they will work out is essential.
Number of words targeted: 15
</sentence>
<assistant_response id="good-example-2">
Analyses of thousands of careers show accomplishment arises from bold, prepared self-direction, not optimistic gamble-taking.
</assistant_response>

<sentence id="good-example-3">
Sentence: I now feel confident living independently.
Number of words targeted: 20
</sentence>
<assistant_response id="good-example-3">
At this point in my life, I genuinely feel very self-assured about managing everything and living entirely on my own.
</assistant_response>

<sentence id="bad-example-3">
Sentence: I now feel confident living independently.
Number of words targeted: 20
</sentence>
<assistant_response id="bad-example-3">
I now feel confident in my ability to manage everything financially and live independently in my dream city.
</assistant_response>
This is a bad example because the original sentence does not talk about financial management and "dream city", while the response does. It adds NEW information, which is unacceptable.

<sentence id="bad-example-4">
Sentence: My uncanny capacity to recall every television channel while forgetting textbook passages prompted my teachers to suspect a specific reading disorder, and subsequent evaluations showed that my academic results were solid rather than disastrous, contradicting the claim that insufficient studying caused my performance.
Number of words targeted: 20
<assistant_response id="bad-example-4">
My recall of TV channels while forgetting textbooks made them suspect a reading disorder until evaluations showed solid grades, debunking insufficient-study claims.
</assistant_response>
This is a bad example because the original noun "teachers" was removed after the rewritten work. The pronoun "them" is unresolved in the response, which is unacceptable.

<sentence id="bad-example-5">
Sentence: My cousin borrowed the headphones after hearing my consistently positive comments.
Number of words targeted: 25
<assistant_response id="bad-example-5">
A considerate roommate is extremely important for the quality of our life at school.
</assistant_response>
This is a bad example because the rewritten sentence contains 14 words, which is not within the permitted range of 23-27. It is thus unacceptable.

**User Prompt:**
Sentence: {original_sentence}
Number of words targeted: {target_length}

Table 10: Nine-shot Mulitple-choice Prompt for CDCP

**User Prompt:**
The following are multiple choice questions. Answer with "the answer is (X)" where X is the correct letter choice.

## Question
Sentence 1: Consumers should not have to pay additional costs for cell phone calls, texting, or voice messaging related to collections.
Sentence 2: Debt collectors and creditors must cease all calls and texts to any device the debtor identifies as not providing free communications.
Which of the following choices is the premise best supporting Sentence 1?
(A) The greater the costs imposed on a debtor, the less money that debtor retains for paying creditors.
(B) Persistent debt collection phone calls may mute or postpone critical emergency notifications on smartphones, thus jeopardizing public safety.
(C) Each unsolicited call or text reaching a device with paid communications should be kept as evidence of noncompliance.
(D) Collection agencies are required to restrict their collection communications to channels that place no monetary cost on the consumer.
## Answer: (A)

## Question
Sentence 1: The more costs assessed to a debtor, the less money the debtor has to pay creditors.
Sentence 2: Debt collectors and creditors must cease all calls and texts to any device the debtor identifies as not providing free communications.
Which of the following choices is the claim best supported by Sentence 1, while also being the premise best supporting Sentence 2?
(A) Consumers ought not be charged extra fees for collection-related cell phone calls, text messages, or voice mails.
(B) The debtor shall promptly provide a written notice that enumerates every phone number on which charges are incurred.
(C) Frequent unwanted calls to pay-per-use mobile phones may obstruct crucial emergency messages that need to reach the debtor.
(D) Before distributing any payments to creditors, court fees and collection charges are subtracted from the debtor's already limited funds.
## Answer: (A)

## Question
Sentence 1: The more costs assessed to a debtor, the less money the debtor has to pay creditors.
Sentence 2: Consumers should not have to pay additional costs for cell phone calls, texting, or voice messaging related to collections.
Which of the following choices is the claim best supported by Sentence 2?(A) Reducing supplementary fees that are imposed on debtors ultimately maximizes the financial resources they can allocate toward repaying their original obligations.(B) Fees levied by the court and statutory penalties usually receive payment priority over creditor claims, instantly shrinking funds remaining for unpaid debts.(C) Creditors and debt collectors must stop all calls and texts to any device the debtor designates as lacking free communication.
(D) Because each extra fee cuts creditors' recovery, they already aim to reduce collection call and message costs, so banning charge passing is unjustified.
## Answer: (C)

## Question
Sentence 1: The Fair Debt Collection Practices Act needs to be updated for modern times.
Sentence 2: Electronic communication is the preferred method of communication for consumers without a doubt.
Which of the following choices is the premise best supporting both Sentence 1 and Sentence 2?
(A) Congress should propose a law that expressly permits debt collectors to interact with consumers through email, text messages, and additional approved electronic communication methods nationwide.
(B) In general, the majority of consumers would unquestionably prefer receiving an email or text message instead of being contacted through a traditional phone call.
(C) Widespread consumer preference for electronic communication demonstrates that debt collectors already interact effectively within the current statutory framework, making revision of the Fair Debt Collection Practices Act unwarranted.
(D) Recognizing that the Fair Debt Collection Practices Act remains rooted in a pre-internet era demonstrates that communication habits are too varied to assert consumers prefer electronic messages.
## Answer: (B)

## Question
Sentence 1: Most consumers would certainly prefer to receive an email or text rather than a phone call.
Sentence 2: Electronic communication is the preferred method of communication for consumers without a doubt.
Which of the following choices is the claim best supported by Sentence 1?
(A) Ongoing staffing deficits and logistical bottlenecks have rendered traditional postal delivery progressively more unreliable and slow.
(B) The Fair Debt Collection Practices Act now requires updating to keep pace with contemporary societal realities.
(C) Digital communications furnish a written record consumers can readily store and consult should any misunderstandings emerge.
(D) Texts and emails let recipients read and respond when convenient, avoiding interruption of their ongoing activities.
## Answer: (B)

## Question
Sentence 1: Most consumers would certainly prefer to receive an email or text rather than a phone call.
Sentence 2: The Fair Debt Collection Practices Act needs to be updated for modern times.
Which of the following choices is the claim best supported by Sentence 1?
(A) Electronic written records, including emails and texts, create an easily searchable trail that consumers might require later as reference during disputes.
(B) Swift progress in artificial intelligence now lets debt collectors deploy automated chatbots and algorithmic dialers that lawmakers in 1977 could never have envisioned.
(C) Undeniably, consumers overwhelmingly favor electronic channels as their primary means of staying in touch and relaying information over alternative approaches.
(D) Emails and text messages let recipients examine information whenever convenient and spare them the sudden intrusion of a ringing phone.
## Answer: (C)

## Question
Sentence 1: If a problem arises with the representative, the customer can review the recording to reveal the truth.

Sentence 2: Automated dialing systems include many built-in controls that protect consumers.
Which of the following choices is the premise best supporting Sentence 2?
(A) Having direct access to verifiable evidence weakens the notion consumers chiefly depend on automated dialer safeguards for protection.
(B) Automated dialers may use filters limiting calls to customers by location, time of day, or number of prior attempts.
(C) Regulatory authorities ought to allow compliant companies to keep using their automated dialing systems without interruption.
(D) Community college consumer rights courses should contain a module that explains the functioning of these built-in controls.
## Answer: (B)

## Question
Sentence 1: Automated dialers can incorporate filters that restrict calls to customers by location, time of day, or the number of prior attempts.
Sentence 2: Automated dialing systems include many built-in controls that protect consumers.
Which of the following choices is the premise best supporting Sentence 2?
(A) A university ethics board approved a comprehensive study testing whether elderly patients feel comfortable receiving medication reminders via automated dialers with built-in consumer protections.
(B) Because automated dialers' flexibility permits adding location, time-of-day, and attempt filters, aggressive collectors can likewise disable them, leaving consumers largely unprotected.
(C) Regulatory authorities ought to allow businesses to depend on automated dialing technologies when delivering timely notifications and important alerts to their customers.
(D) Should any issue ever arise regarding the representative's conduct, the customer is entitled to examine the recording in order to uncover the actual facts.
## Answer: (D)

## Question
Sentence 1: Automated dialers can incorporate filters that restrict calls to customers by location, time of day, or the number of prior attempts.
Sentence 2: If a problem arises with the representative, the customer can review the recording to reveal the truth.
Which of the following choices is the claim best supported by both Sentence 1 and Sentence 2?
(A) Federal consumer-protection rules impose hefty penalties on firms calling outside permitted hours or surpassing contact quotas.
(B) Automated dialing systems incorporate numerous internal safeguards that are specifically designed to protect consumers from harm.
(C) Recent consumer-protection regulations impose limits on call frequency and require that verification of conversations be recorded.
(D) Every customer service call is automatically recorded and stored in a secure database for ninety days.
## Answer: (B)

## Question
Sentence 1: {first_context_sentence}
Sentence 2: {second_context_sentence}
{question}
(A) {choice_A}
(B) {choice_B}
(C) {choice_C}
(D) {choice_D}
## Answer

