# OpenReview forum: "ArgQA: Evaluation of Reasoning Over Elementary Logical Structures in Arguments"
_ICLR.cc/2026/Conference — ICLR 2026 Conference Withdrawn Submission_

### Official Review · Reviewer_XSW5 · 2025-10-29

**Soundness:** 3
**Presentation:** 4
**Contribution:** 3
**Rating:** 6
**Confidence:** 3

**Summary:**

This paper introduces ARGQA, a novel benchmark dataset designed to evaluate the logical reasoning capabilities of LLMs on authentic, real-world arguments. The authors argue that existing benchmarks are often inadequate, as they either rely on synthetically generated text or exam-style questions that use unnatural language, limiting their real-world applicability. ARGQA consists of 3,807 multiple-choice questions sourced from arguments across four distinct domains: product reviews, argumentative essays, e-rulemaking comments, and medical research abstracts. The dataset is specifically designed to assess an LLM's ability to recognize and reconstruct three elementary logical structures: linear, convergent, and divergent. The questions are standardized into nine distinct types, each corresponding to a specific logical relationship between the context sentences and the potential answers modeled by a graph, with correct alternative MCQ options constructed with the graph. Experiments on several modern LLMs (including the Mistral, Llama, and GPT series) demonstrate that this task remains challenging for modern models. The strongest model, GPT-03, achieved an accuracy of only 61.81% in a 9-shot setting.

**Strengths:**

Novel Problem Formulation: The paper identifies a clear and important gap in current LLM evaluation: the lack of benchmarks that test structural logical reasoning on natural, authentic arguments, constructed with consideration of standard logic/argument structures in philosophy.

Thoughtful Dataset Construction: The methodology and particularly the explanation for creating ARGQA is strong/ It is well-defended and transparently explained.

Fine-Grained Analysis: The paper's design, which is built around three elementary structures and nine specific question types, is a key strength. This allows for a better analysis than a single accuracy score, as demonstrated by the paper's ability to break down performance by structure type and logical role.

Clear Writing: The paper is well-written and the motivation is clearly established with a solid theoretical grounding in argumentation theory.

**Weaknesses:**

The paper's primary weakness lies in a few missed opportunities for a more comprehensive evaluation, which are mostly captured in the questions below.

The evaluation is limited to 0-shot and 9-shot accuracy using greedy decoding. This leaves open questions about model consistency, the effects of different sampling strategies, and the potential for few-shot learning with more examples.

The paper does not fully explore the robustness of the models. For example, it is unclear how the models perform compared to human paraphrasing of the propositions or if their performance is brittle/varied across models.

**Questions:**

How does model performance on ARGQA compare to performance on equivalent, purely symbolic logic problems (e.g., "A supports B, B supports C")?

Can you provide more evidence for the claim that "linked structures" are infrequent in practice? Was their exclusion also motivated by the difficulty of data collection?

Was the evaluation of more complex, composite logical structures (beyond the three elementary ones) considered as a next step?

Could you provide an example of how "the lack of explicit context often obscures the logic" (Section 3.3) and clarify if this implies some questions are ambiguous without external knowledge?

What was the rationale for choosing 0-shot and 9-shot settings? Since 9-shot provides only one example per question type, how does performance scale with more in-context examples?

---

### Official Review · Reviewer_n2NV · 2025-10-30

**Soundness:** 2
**Presentation:** 2
**Contribution:** 2
**Rating:** 4
**Confidence:** 3

**Summary:**

This paper aims to evaluate the capability of LLMs in processing the basic logical structures in real-world text. It is claimed to overcome the problem of lack of natural language in existing datasets.

In this work, the authors focuses three types of basic logical structures, and then define 9 types of questions based on them by introducing some distractors into the three basic logical structures.

The question generation starts from source selection. Several datasets with desired logical labels are chosen. Then correct triplets are extracted from the text source and refined by paraphrasing. Distractors are then introduced to ad some noise. Finally, the questions are designed to query the logical relationship between the context and correct options.

**Strengths:**

This paper targets the reasoning capability of LLMs, and specifically focuses on the capability to analyze over the basic logical structures, which is a good intention. In addition, the work pointed out some unnatural problem in existing benchmark, which should be an important aspect of the research in this direction.

**Weaknesses:**

1. The constructed questions only contain triplets extracted from original datasets with natural language usage, then the content is also not the original natural content we could encounter in daily life. In addition, the questions are simply determine the relationship between the context and the options, which does not look like a natural scenario either.

2. Some concepts are not clearly defined before usage. For example, which propositions are context propositions in the three types of logical structures focused in this work? I was guessing that it should be the first two propositions in linear and convergent structures and the first one proposition in the divergent structure. However, around line 233, it is written '.... one or both of the context propositions ...', which seems to imply that each structure has exactly two context propositions.

3. The explanation on the logical distractor is not clear enough. More concrete examples is needed to illustrate this.

4. The distractors are generated by GPT-o3, while there seems to be no approach to validate and ensure the correctness.

**Questions:**

1. Although there are unnatural questions in the existing benchmarks, are there also a sufficient amount of questions following natural expression? It seems possible that existing benchmarks contain both natural and unnatural expressions.

2. The categorization on the logical structures does not follow common logical analysis frameworks, e.g. prepositional logic or first-order logic. How to justify that the adopted three type of logical structures are reasonable enough for evaluating the reasoning capability of LLMs. In addition, does this covers all possible logical structures?

3. The selected source datasets already contain the relation annotations. Then it seems that the target logical structures are already available in these datasets, and what is the unique contribution of this work?

4. The constructed questions only contain triplets extracted from original datasets with natural language usage, then the content is also not the original natural content we could encounter in daily life. How would the authors interpret this?

5. How are the direct graph generated from the text, by LLM? It seems that sometimes the logical structure in the text could be complex or vague, which cause difficulty for extracting clear logical structure.

---

### Official Review · Reviewer_MwTb · 2025-10-30

**Soundness:** 2
**Presentation:** 3
**Contribution:** 2
**Rating:** 4
**Confidence:** 4

**Summary:**

This paper introduces ARGQA, a new benchmark dataset designed to evaluate the logical reasoning capabilities of Large Language Models (LLMs) over elementary logical structures—namely linear, convergent, and divergent argument structures. The dataset contains 3,807 multiple-choice questions derived from real-world arguments across four domains: product reviews, argumentative essays, e-rulemaking comments, and medical research abstracts. Each question is crafted to assess the model’s ability to recognize or reconstruct a specific logical structure, with carefully designed distractors that reflect common logical misunderstandings.

**Strengths:**

1) ARGQA is built from authentic arguments across multiple domains, which makes it more representative of real-world reasoning tasks than synthetic or exam-style benchmarks.
2) The dataset is carefully designed around three elementary logical structures (linear, convergent, divergent), each with three question types.
3) The paper provides a detailed analysis of model performance across logical structures and domains.

**Weaknesses:**

1) While the paper focuses on three elementary structures, it excludes linked structures and more complex argument forms. This limits the benchmark’s ability to assess higher-order reasoning or argumentative dynamics, which are common in real-world discourse. The exclusion is justified by practical concerns, but the dataset still falls short of capturing the full complexity of argumentation.
2)  Authors argue that "However, both synthetic and exam-style questions contain unnatural language, thereby limiting their applicability to real-world contexts." But actually,  unnatural language does not limit the applicability in the real-world context, as there are many scenarios using unnatural language, e.g., symbolic rules. Also, the dataset proposed by the authors is not more applicable than others.
3) The use of multiple-choice questions (MCQs) simplifies evaluation but may not fully reflect reasoning depth. MCQs can be prone to guessing, surface pattern matching, or elimination strategies that do not require genuine understanding. A more open-ended format (e.g., argument reconstruction or generation) could provide a richer assessment of reasoning capabilities.
4) The paper assumes transitivity of support relations, which may not always hold in natural language argumentation. This assumption can oversimplify the logical structure and may lead to misclassification of valid arguments or distractors. A more nuanced treatment of support would better reflect real-world reasoning.

**Questions:**

In natural language, support is often contextual, defeasible, or non-transitive. Could this assumption lead to misclassification of valid arguments or distractors? Have you evaluated cases where transitivity breaks down?

---

### Official Review · Reviewer_v3xo · 2025-11-10

**Soundness:** 2
**Presentation:** 2
**Contribution:** 3
**Rating:** 4
**Confidence:** 4

**Summary:**

This paper introduces ArgQA, a benchmark designed to evaluate logical reasoning. The authors argue that existing benchmarks are often limited because they either rely on automatic conversion of symbolic logic into natural language or use curated questions from standardized exams such as the LSAT. In contrast, ArgQA contains authentic arguments drawn from four domains and is designed to assess different reasoning structures, including linear, convergent, and divergent forms. Using this dataset, the authors evaluate the performance of Qwen and GPT-o3, concluding that both models show significant limitations in logical reasoning.

**Strengths:**

1.The motivation for creating this dataset is clear and meaningful.

2.The experimental analysis is broad and well designed.

**Weaknesses:**

1.Figure 3 is difficult to interpret. How is the diagram constructed, and how does it relate to real-world reasoning scenarios?

2.Several details in Step 2 need clarification. Do the triplets correspond to S1, S2, and the ground truth proposition? You mention that directed graphs are extracted from annotations: are these annotations provided by the source datasets or generated manually, and through what process?

3.In Step 5, how do you ensure that GPT-3o actually produces distracting choices rather than arbitrary alternatives?

**Questions:**

Not a question, but the reviewer found table 1 is very similar to the table in [1], however this paper is not even been discussed or cited.

[1]. Towards LogiGLUE: A Brief Survey and A Benchmark for Analyzing Logical Reasoning Capabilities of Language Models

---

### Note · Authors · 2025-11-15

**Comment:**

Thanks for the effort everyone has put into reviewing this work

**Withdrawal Confirmation:**

I have read and agree with the venue's withdrawal policy on behalf of myself and my co-authors.